# Pre-Amplification of Cell-Free DNA: Balancing Amplification Errors with Enhanced Sensitivity

**DOI:** 10.3390/biom15060883

**Published:** 2025-06-17

**Authors:** Wei Yen Chan, Ashleigh Stewart, Russell J. Diefenbach, Elin S. Gray, Jenny H. Lee, Richard A. Scolyer, Georgina V. Long, Helen Rizos

**Affiliations:** 1Faculty of Medicine, Health and Human Sciences, Macquarie University, Sydney, NSW 2109, Australia; weiyen.chan@hdr.mq.edu.au (W.Y.C.); ashleigh.stewart@mq.edu.au (A.S.); russell.diefenbach@mq.edu.au (R.J.D.); jenny.lee@mq.edu.au (J.H.L.); 2Melanoma Institute of Australia, The University of Sydney, Sydney, NSW 2065, Australia; richard.scolyer@health.nsw.gov.au (R.A.S.); georgina.long@sydney.edu.au (G.V.L.); 3Centre for Precision Health, Edith Cowan University, Joondalup, WA 6027, Australia; e.gray@ecu.edu.au; 4School of Medical and Health Sciences, Edith Cowan University, Joondalup, WA 6027, Australia; 5Department of Medical Oncology, Chris O’Brien Lifehouse, Sydney, NSW 2050, Australia; 6Faculty of Medicine and Health, The University of Sydney, Sydney, NSW 2065, Australia; 7Charles Perkins Centre, The University of Sydney, Sydney, NSW 2006, Australia; 8Tissue Pathology and Diagnostic Oncology, Royal Prince Alfred Hospital & NSW Health Pathology, Sydney, NSW 2006, Australia; 9Department of Medical Oncology, Royal North Shore and Mater Hospitals, Sydney, NSW 2060, Australia

**Keywords:** circulating tumour DNA, melanoma, detection sensitivity

## Abstract

Circulating tumour DNA (ctDNA) is a promising biomarker for personalised oncology. However, its clinical utility is limited by detection sensitivity, particularly in early-stage disease. T-Oligo Primed Polymerase Chain Reaction (TOP-PCR) is a commercial amplification approach utilising an efficient “half-adapter” ligation design and a single-primer-based PCR strategy. This study evaluated the clinical value and application of cell-free DNA (cfDNA) pre-amplification. cfDNA amplification with TOP-PCR preserved DNA size profiles and resulted in a 22 bp size increase due to the half-adaptor ligation. Gene target amplification rates varied, showing lower efficiency for the GC-rich TERT promoter amplicon and higher efficiency for the *BRAF* and *TP53* amplicons. Optimised pre-amplification (20 ng cfDNA input and 5–7 cycles of PCR) enhanced ctDNA detection sensitivity and expanded sample availability for the detection of multiple tumour-informed mutations. Importantly, PCR errors emerged in pre-amplified cfDNA samples, underscoring the necessity for negative controls and the establishment of stringent mutation positivity thresholds.

## 1. Introduction

Circulating tumour DNA (ctDNA) is a promising biomarker for personalised medicine in oncology. The analysis of ctDNA offers real-time, longitudinal information regarding tumour dynamics, and has proven valuable for detecting minimal residual disease and monitoring treatment response and resistance [1,2,3]. However, detecting ctDNA remains challenging as only a small fraction of circulating cell-free DNA (cfDNA) is tumour-derived, particularly in early-stage disease or after curative-intent treatment, where the mutant allele fraction is often less than 0.1% [4,5].

Recent efforts to improve the sensitivity of ctDNA detection have focused on enhancing the simultaneous identification of 10–100 s of tumour-associated variants with molecular barcoding-based next-generation sequencing (NGS) [6,7,8]. Achieving high sensitivity and specificity can be achieved by increasing the depth of cfDNA sequencing, extending sequencing to the whole genome, and applying robust bioinformatic pipelines to suppress background errors and to distinguish true tumour mutations from non-tumour-related somatic variants [9,10]. An alternative approach involves digital PCR-based detection of multiple tumour-informed mutations in pre-amplified cfDNA. T-Oligo-Primed Polymerase Chain Reaction (TOP-PCR) utilises a single small linear adaptor for enhanced ligation and enables the efficient non-selective amplification of cfDNA [11].

In this study, the real-world clinical value of TOP-PCR pre-amplification for ctDNA analysis was examined. This approach offers enhanced ctDNA detection sensitivity, which is valuable in clinical practice, although errors introduced during pre-amplification can compromise specificity.

## 2. Methods and Materials

### 2.1. Sample Collection and cfDNA Extraction and Quantification

Patients with stage III or IV melanoma and treated with a single agent, i.e., anti-PD-1, or a combination therapy of anti-PD-1 and anti-CTLA-4 immune checkpoint inhibitor were included in this study (Appendix A). Blood samples were collected before treatment or early during the treatment (6–12 weeks from the start of treatment) using 10 mL EDTA vacutainer tubes (Becton Dickinson, Franklin Lakes, NJ, USA).

This study also utilised cfDNA extracted from a human melanoma or human dermal fibroblast (HDF1314, Cell Applications, San Diego, CA, USA) cell culture medium after 3–7 days in culture [12,13,14]. Blood and culture medium were processed within four hours of collection by centrifugation at 800× *g* for 15 min, followed by a second centrifugation at 1600× *g* for 10 min. Double spun plasma and medium were stored at −80 °C until extraction.

cfDNA was extracted from 2–4 mL of plasma or cell culture medium using QIAamp^®^ Circulating Nucleic Acid Kit (Qiagen, Hilden, Germany) as per manufacturer’s instructions. Samples were eluted in 100 µL sterile distilled water. Cell-free DNA was quantified using the Qubit High Sensitivity dsDNA Kit and the Qubit 3.0 Fluorometer (Thermo Fisher Scientific, Waltham, MA, USA), and cfDNA profiles were analysed using Cell-free DNA ScreenTape and TapeStation 4150 (Agilent Technologies, Santa Clara, CA, USA).

### 2.2. Pre-Amplification Using T-Oligo Primed PCR

A range of 0.5 to 20.0 ng of cfDNA (0.5, 1.25, 2.5, 5.0, 10.0, 11.25, 20 ng) in a volume of 6.6 µL was amplified using the DNA TOP-PCR Kit, as described by the manufacturer (D01, Top Science Biotechnologies Inc., Taiwan, China). TOP-PCR is a three-step process consisting of (i) the end repair and A-tailing of the DNA, (ii) the ligation of half-adaptors to the DNA, and (iii) PCR amplification using only the T-oligo primer to selectively amplify ligated DNA [11]. When required, input cfDNA was concentrated using Eppendorf Concentrator Plus (Eppendorf, Hamburg, Germany) to ≥3 ng/µL. Ligated cfDNA was amplified for 4–15 cycles and eluted in 30 µL sterile-distilled water following purification with AMPure XP beads as per manufacturer instructions, except that the ratio of beads to sample was approximately 1.93 (Beckman Coulter, Brea, CA, USA). Amplification efficiency was assessed by quantifying DNA yields using Qubit, with efficiency determined using the formula Nf = No (1 + Y)^n^
^–^
^1^, where No and Nf are the initial and final DNA copy numbers, n is the number of PCR cycles, and Y is the efficiency of primer extension per cycle [15].

### 2.3. Droplet Digital PCR

The ddPCR reaction mix was prepared as per manufacturer instructions using ddPCR Supermix for Probes (no dUTP; Bio-Rad Laboratories, Berkeley, CA, USA) with wild-type and mutant probes (FAM/HEX; Appendix A) included in each assay. Droplets were generated using the QX200 AutoDG (Bio-Rad Laboratories) and cycled on the C1000 Touch Thermocycler (Bio-Rad Laboratories). Droplets were analysed using the QX600 Droplet Reader and QX Manager 2.0 (Bio-Rad Laboratories). Each ddPCR run included (i) a no-template control, (ii) a positive control (a known positive sample, or a mutant gblock-spiked HDF1314 human dermal fibroblast cfDNA sample), and (iii) a negative control (HDF1314 human dermal fibroblast cfDNA). Identical manual thresholds were applied across all samples within a ddPCR run.

### 2.4. Statistics

For statistical analysis, GraphPad Prism software v.10 was used. Figure legends specify the statistical analysis used and define error bars.

## 3. Results

### 3.1. Performance and Optimisation of TOP-PCR cfDNA Pre-Amplification

The performance of TOP-PCR was validated using 0.5 to 20 ng input cfDNA derived from melanoma patients. Ligated cfDNA was amplified for 15 cycles as recommended by the manufacturer. The TOP-PCR yield was highly variable (range: 443–1237 ng) and inversely correlated with the initial cfDNA input (Pearson correlation, r = −0.9027, *p* = 0.0054; Figure 1A). The diminishing TOP-PCR yield with increasing input cfDNA appears to reflect suboptimal efficiency potential due to the early saturation of reaction components [15,16]. By reducing the PCR cycle number from 15 to 5 cycles, the end-product yield maintained a linear increase relative to cfDNA input (Pearson correlation, r = 0.9882, *p* < 0.001; Figure 1B).

Next, we aimed to establish robust amplification conditions for 20 ng of input cfDNA, which is equivalent to approximately 6000 copies of the haploid human genome [17]. This DNA quantity allows for rare variant detection with a limit of detection as low as 0.02% (i.e., 1 in 6000). The efficiency of TOP-PCR with 20 ng input cfDNA was evaluated across four to seven amplification cycles and compared with that of 5 ng input cfDNA. We achieved the near doubling of DNA product during each amplification cycle over the seven cycles with 5 ng of input cfDNA, and a consistent but a slightly lower amplification efficiency with 20 ng of input cfDNA (Figure 1C). These yields, as measured by Qubit, corresponded to a PCR efficiency of 90% per cycle for 20 ng of input cfDNA and 116% per cycle for 5 ng of input cfDNA. Based on these data, we selected five and seven TOP-PCR amplification cycles for our downstream analyses, balancing amplification yield while minimizing this risk of artefactual variant calls.

The size profile of unamplified and TOP-PCR-amplified cfDNA was also evaluated. Unamplified cfDNA derived from melanoma patients exhibited a mono-nucleosomal DNA median peak size of 193 bp (range: 183–205; *n* = 21 stage IV melanoma cfDNA samples) (Figure 2). After adaptor ligation and TOP-PCR amplification of these patient samples, the mono-nucleosomal DNA peak increased to an expected peak size of 208 bp (range: 195–218; median unligated peak size = 193 + 22 = 215 bp expected product; Figure 2A,B). Similar to Nai et al. [11], we noted that TOP-PCR accentuated the di-nucleosomal cfDNA peak (Figure 2B). Specifically, the amount of di-nucleosomal DNA accounted for a median of 36.1% (range: 29.6–43.7%; *n* = 21 samples) of TOP-PCR-amplified cfDNA (100–700 bp), compared to a median of 12.1% (range: 8.6–17.7%; *n* = 21 samples) of unamplified cfDNA (100–700 bp; Figure 2C; *p*-value < 0.001). Importantly, over 90% (range: 86–97%) of the TOP-PCR cfDNA product was within 100–700 bp in length, representing a significant increase (*p*-value < 0.001) from the unamplified cfDNA, where only 86% (range: 74–96%) was within the 100–700 bp range (Figure 2D).

Finally, although the yield of TOP-PCR-amplified cfDNA from 20 ng of input cfDNA was variable after seven PCR cycles, the yield strongly correlated with cfDNA fragmentation patterns. PCR amplification yields increased as the proportion of DNA within the 100–700 bp size range and the proportion of mono-nucleosomal and di-nucleosomal cfDNA increased (Figure 2E). Collectively, these data indicate that, while both di-nucleosomal and mono-nucleosomal DNA inputs correlate with amplification efficiency, the di-nucleosomal DNA is more efficiently amplified during TOP-PCR (Figure 2C).

### 3.2. TOP-PCR Pre-Amplification Efficiency Varies Across Cancer Gene Targets

To examine the sequence-specific impact of TOP-PCR cfDNA pre-amplification, we examined the end-point yield (at PCR saturation, 15 cycles) of five common melanoma-associated gene targets (Appendix A). The gene-specific yields of wild-type *BRAF*, *NRAS*, *CDKN2A*, *TERT*, and *TP53* were determined using ddPCR after TOP-PCR pre-amplification. As shown in Figure 3, *TP53* exhibited the highest fold increase relative to unamplified cfDNA (238-fold increase in DNA copies), followed by *CDKN2A* (109-fold), *BRAF* (28-fold), *NRAS* (18-fold), and *TERT* (1.3-fold). We examined the impact of molecular enhancers such as betaine or DMSO to try and improve the consistency of target amplicon amplification. These small-molecule additives did not notably improve the amplification yield of *TP53*, *CDKN2A*, *BRAF*, or *NRAS*; instead, they reduced the yield of *TP53*, *BRAF*, and *NRAS*. In contrast, the addition of betaine and DMSO consistently, albeit marginally, increased the end-product yield of *TERT* from 1.3-fold to 3.0-fold and 2.7-fold in the standard betaine- and DMSO-treated PCR reactions, respectively (Figure 3).

### 3.3. TOP-PCR Pre-Amplification Increases ctDNA Testing and Detection Sensitivity

A potential advantage of cfDNA pre-amplification is the increased availability of cfDNA, which can often fall below 20 ng/mL plasma in melanoma patients [18]. Increased cfDNA amounts enable testing for multiple tumour-associated mutations and minimise subsampling errors, ensuring the more representative and comprehensive detection of mutations.

To assess the clinical utility of TOP-PCR, we examined the detection of tumour-associated variants in a small cohort of stage III and IV melanoma patients undergoing immune checkpoint inhibitor therapy (Appendix A). In the advanced melanoma cohort, cfDNA was collected early during the course of systemic therapy (mean: 8.8; range: 6–12 weeks after therapy initiation). For each patient, 20 ng of cfDNA was amplified, and 40 ng of the resulting amplified cfDNA was used to analyse two tumour-associated variants per patient sample in independent ddPCR (Table 1). This approach highlighted two key advantages of TOP-PCR: (i) increased sample availability to screen for multiple variants, overcoming the challenge that not all tumour-associated mutations are detectable in circulation, and (ii) increased extremely-low-mutation droplet counts, enhancing detection confidence (Appendix A). For instance, patient 52318 had an undetectable *TERT* promoter mutation (Mutation 1: *TERT* c.-146C>T), but had a detectable *BRAF* D594N mutation (Mutation 2: c.1780G>A; Table 1). Importantly, ctDNA was detectable for at least one tumour-associated variant in all seven patients who progressed on immune checkpoint inhibitor therapy (median time to progression: 7 months; range: 1–58 months; Table 1).

We also evaluated the effectiveness of TOP-PCR in enhancing the sensitivity of ctDNA detectability in earlier-stage melanoma (stage III; Appendix A). In all patients, TOP-PCR increased ctDNA mutation droplet counts, and in patients 45141 and 48810, ctDNA was undetectable in the unamplified cfDNA samples but became detectable after TOP-PCR amplification. These findings highlight the value of TOP-PCR pre-amplification in increasing the cfDNA yield and detection sensitivity, reducing subsampling errors and facilitating multi-mutation tracking to improve the detection of clinically significant mutations.

### 3.4. TOP-PCR-Errors and Impact on Rare Variant Detection

Residual amplification errors from TOP-PCR may impact the detection of rare tumour-associated mutations. To monitor PCR errors, we included multiple negative control reactions for each mutation assay. Due to the significant quantities of control DNA required, we obtained control wild-type cfDNA from cultured primary neonatal HDF1314 human dermal fibroblasts. Negative control reactions were performed with and without pre-amplification (at both five and seven amplification cycles), with an average of four ddPCR runs per probe. ddPCR positivity thresholds were derived based on the number of FAM+/HEX- mutant droplets in negative-control PCR reactions. Only FAM+/HEX- quadrant droplets were classified as mutant droplets as several BioRad wet lab validated probes, such as *TP53*R282W (Appendix A), generate false-positive FAM+/HEX+ droplets.

Without TOP-PCR amplification, no mutant-positive droplets were detected in over 110 ddPCR runs using HDF1314-derived cfDNA, across 20 tested mutations (Appendix A, Appendix A). Of these 20 mutations, 18 were subsequently tested following TOP-PCR amplification and presented in Appendix A. After five cycles of TOP-PCR amplification, 5 out of 51 (9.8%) ddPCR runs generated 1–2 false-positive droplets, occurring specifically with the *BRAFV*600E and *BRAFV*600K probes (Appendix A). In contrast, the false-positive rate was significantly higher after seven TOP-PCR amplification cycles. False-positive droplets (1–6 false-positive droplets) were detected in 19 out of 58 (33%) ddPCR reactions and were detected for multiple ddPCR wet lab validated assays (*BRAF*V600E, *NRAS*Q61K, *NRAS*Q61L, *IDH1*R132H, *TP53*R282W, and *TERT* C228T; Appendix A).

## 4. Discussion

Detecting tumour-associated variants in cfDNA remains challenging as ctDNA can account for less than 0.01% of the total cfDNA, particularly in early-stage disease [19,20]. Additionally, some cancer types, including prostate, kidney, and tumours located in the central nervous system, release low levels of ctDNA into the circulation [21,22,23]. To improve the detection limit of ctDNA, recent approaches rely on NGS strategies to amplify and sequence tumour-associated mutations in a single assay [10,24]. The sensitivity of ctDNA detection increases as the number of analysed mutations and the number of unique cfDNA molecules sequenced increase [19]. While these approaches offer high sensitivity, they are also limited by cost, extended turnaround times, and the need for complex bioinformation analysis pipelines to suppress background errors and identify true tumour-associated mutations [25]. Furthermore, the library preparation and capture steps introduce DNA errors via oxidative damage and strand bias in sequencing can limit mutation detection [26].

In this study, we examined the value and limitations of incorporating a simple and rapid ligation pre-amplification step into a standardised ddPCR protocol for detecting tumour-associated variants in cfDNA. TOP-PCR relies on the TA ligation of a single, 11 bp adaptor, which is reportedly more efficient than the ligation of the Illumina loop and Y adaptors [11]. The entire process from cfDNA extraction to ddPCR mutation detection took place in less than 16 h, which is particularly critical in clinical contexts where timely turnaround of patient results can inform treatment decisions.

TOP-PCR preserved the cfDNA size profile and effectively amplified, albeit unevenly, both mono- and di-nucleosomal cfDNA fragments. The relative enrichment of di-nucleosomal fragments may reflect their reduced sensitivity to nuclease degradation, as shorter cfDNA fragments are less tightly maintained on the nucleosome and are more prone to nicking and degradation [27]. Consequently, di-nucleosomal cfDNA may be more efficiently ligated and amplified during the TOP-PCR process. The enrichment of di-nucleosomal cfDNA could potentially lead to the loss of representation of rare tumour variants, as ctDNA is generally of a smaller size compared to normal cell cfDNA [28]. To optimise assay performance, we undertook an iterative evaluation of the number of PCR cycles used during pre-amplification. Initial experiments using seven cycles increased the overall cfDNA yield, which is beneficial for samples with limited input. However, this was accompanied by a higher rate of false-positive mutation calls, likely due to polymerase-induced errors. Based on these data, five amplification cycles were subsequently tested and provided the optimal balance—maintaining high specificity while still enhancing sensitivity. Another strategy to mitigate amplification-associated errors is to perform replicate, independent pre-amplification reactions, if sufficient DNA is available, although this approach increases cost and sample consumption.

The limitations of TOP-PCR are shared with all polymerase-based amplification approaches. First, although TOP-PCR uses a DNA polymerase with high fidelity and low bias (Novagen KOD in [11]), DNA errors are introduced during amplification, necessitating strict control measures. Accordingly, multiple negative controls for each amplicon must be included in every TOP-PCR run. We also recommend establishing experimentally validated thresholds (based on multiple positive and negative control reactions) for calling a true-positive mutation, and the inclusion of at least two independent ddPCR experiments. Our recommended TOP-PCR amplification workflow is shown in Appendix A. Additionally, unbiased calling algorithms that establish assay-specific noise profiles, such as CASTLE, may also prove useful in estimating the true concentration of mutational fragments in a sample [29]. Second, amplification bias occurs during PCR, and not all amplicons are amplified with equal efficiency in the cfDNA population. This was also evident with TOP-PCR. Despite testing the molecular enhancers DMSO and betaine, which are known to improve amplification efficiency in GC-rich regions such as the *TERT* promoter [30], none consistently improved PCR efficiency across all the amplification targets tested in our study. Previous reports have demonstrated that the PCR enhancers Q-solution and 7-deaza-dGTP enhanced the amplification of the *TERT* promoter, and it would be worthwhile testing these additives in TOP-PCR [31,32]. Differences in DNA fragment quality, size distribution, or GC content, as well as variability in adaptor ligation, can lead to the uneven representation of genomic regions in the amplified product [33,34]. Third, the cost of TOP-PCR is not insignificant, adding approximately AUD $105 per sample to ctDNA analysis. However, it remains more affordable than NGS and presents a rapid, cost-effective option for clinical applications.

The principal value of incorporating pre-amplification in ctDNA analyses is that it enables the examination of multiple tumour-informed mutations in cfDNA samples, which is often limited in cancer patients. Our real-world results confirmed that TOP-PCR increased the detection rate of pre-treatment ctDNA in stage III melanoma patients from 30% (12/40) to 48% (19/40), demonstrating its clinical utility [35]. The TOP-PCR process also allows for P5-P7 ligation followed by hybrid capture and next-generation sequencing, although this approach would benefit from the inclusion of unique molecular identifiers. Importantly, TOP-PCR is one of several available pre-amplification techniques, each with distinct advantages and limitations. We do not present TOP-PCR as superior to alternative methods, but rather as a practical and efficient option, particularly well suited in clinical settings that demand rapid turnaround, low-input DNA, and cost-effective workflows. Recent innovations, such as the 1DF-PCR assay (TopScience Biotechnologies), allows for direct amplification from plasma samples without prior DNA extraction, potentially offering further improvements in cost and ctDNA detection, as it eliminates purification-associated loss [36]. Finally, the use of multiplex tumour-mutation platforms (e.g., the BioRad QX600 instrument) now allows for the concurrent analysis of multiple mutations per patient at single-molecule detection limits, making pre-amplification–ddPCR mutation detection pipelines like TOP-PCR increasingly relevant [37,38].

## 5. Conclusions

Incorporating a rapid ligation pre-amplification step into a ddPCR workflow can enhance the sensitivity of detecting tumour-informed gene variants, especially in early-stage cancers with low ctDNA levels. TOP-PCR pre-amplification increases sample yield and enables multi-probe tracking, providing a cost-effective alternative to next-generation sequencing. However, this approach has significant limitations—most notably, the risk of introducing of PCR errors—which must be carefully monitored. When appropriately controlled, cfDNA pre-amplification can improve ctDNA detection to support clinical decision-making.

## Figures and Tables

**Figure 1 biomolecules-15-00883-f001:**
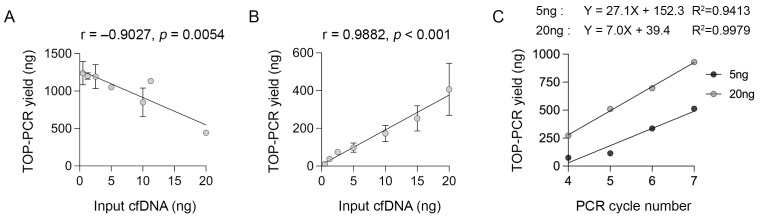
The performance of TOP-PCR. The total cfDNA yield post-TOP-PCR amplification is shown relative to the initial cfDNA PCR input after (**A**) 15 cycles of amplification (at least two independent amplifications per input cfDNA) and (**B**) 5 cycles of amplification (*n* = 1 for <5 ng cfDNA input, at least two independent amplifications for all other input cfDNA amounts). Pearson correlation coefficient and two-tailed *p*-values are shown for each graph. (**C**) Amplification efficiency was tested using 5 ng and 20 ng inputs and four to seven PCR cycles (*n* = 1 amplification reaction from a dilution series of a single cfDNA sample mix). The regression line equations and goodness-of-fit data (R^2^) are also shown. Data shown are mean ± s.d; ng, nanogram.

**Figure 2 biomolecules-15-00883-f002:**
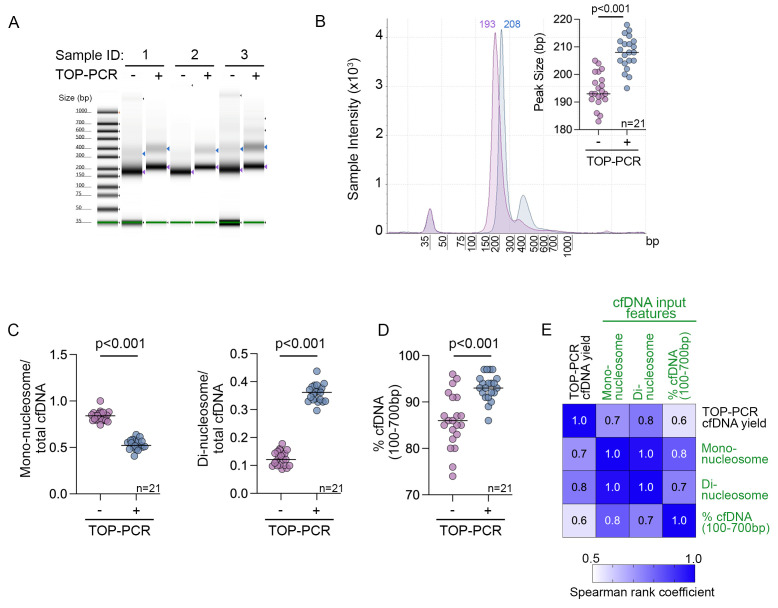
The effect of TOP-PCR on cfDNA mono-nucleosome and di-nucleosome peaks. (**A**) The electrophoresis of melanoma cfDNA pre (-) and post (+) TOP-PCR amplification (20 ng cfDNA input, five cycles of amplification). cfDNA derived from three melanoma patients was separated using the Agilent TapeStation 4150. Purple arrows indicate the mono-nucleosomal DNA, and blue arrows indicate the di-nucleosomal DNA (where visible). The molecular weight ladder (shown on the left) includes two internal standards (lower and upper markers); variation in their appearance across samples is due to image scaling adjustments made for visual display. The TapesStation 4150 electrophoresis without intensity scaling is shown in Appendix A. (**B**) The size profile of representative patient-matched cfDNA pre- and post-TOP-PCR amplification. (Inset) The scatter plot showing the median mono-nucleosome peak size for 21 stage IV melanoma patient-matched cfDNA samples pre- and post-TOP-PCR amplification (20 ng cfDNA input, seven PCR cycles). Median is shown in each graph. Peak size data were compared using Wilcoxon matched-pairs signed rank test. (**C**) The yield (quantitated on TapeStation 4150) of mono-nucleosome (**left**) and di-nucleosomal (**right**) cfDNA pre- and post-TOP-PCR amplification of 20 ng cfDNA input for seven PCR cycles is shown relative to the total cfDNA yield (100–700 bp) in 21 stage IV melanoma patient-matched cfDNA samples. Median is shown in each graph. Data were compared using the Wilcoxon matched-pairs signed-rank test. (**D**) The percentage of DNA between 100 and 700 bp (quantitated on TapeStation 4150) is shown pre- and post-TOP-PCR amplification (20 ng cfDNA input, seven PCR cycles) in 21 stage IV melanoma patient-matched samples. Median is shown in each graph. Data were compared using the Wilcoxon matched-pairs signed-rank test. (**E**) The correlation matrix of DNA yield following TOP-PCR amplification (20 ng input, seven PCR cycles) with cfDNA features including cfDNA mono-nucleosome input, cfDNA di-nucleosome input, and % cfDNA input (100–700 bp DNA fraction). Data derived from 21 stage IV melanoma patients. The Spearman rank correlation coefficients are shown within the matrix. Correlation *p*-values are all <0.01.

**Figure 3 biomolecules-15-00883-f003:**
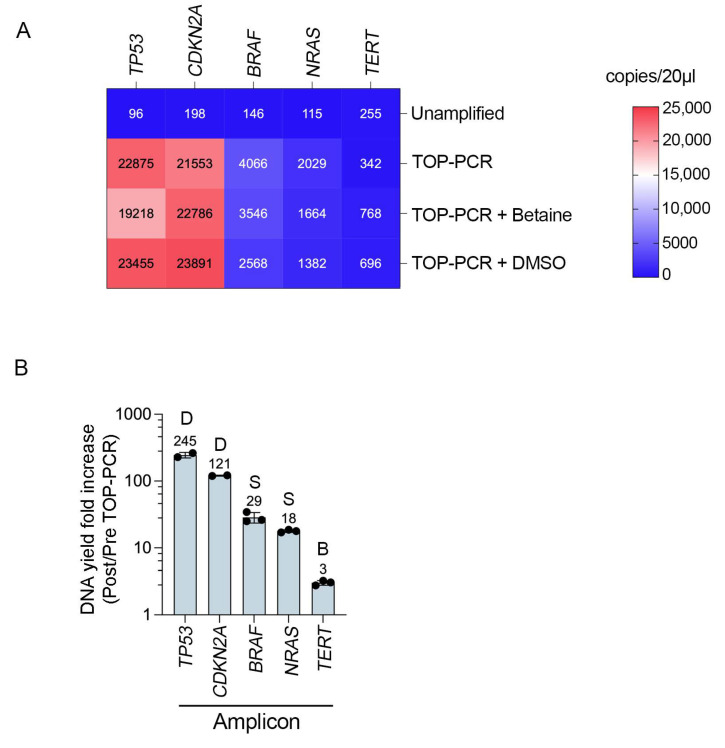
Gene-specific effects on pre-amplification efficacy. (**A**) Heat map showing the yield (copies/20 µL in ddPCR) of indicated gene targets without pre-amplification, or after 15 cycles of TOP-PCR amplification with no molecular enhancers (TOP-PCR) or including 0.5 M betaine or 5% DMSO. The mean DNA yield of 2–3 independent PCR reactions is shown in each cell. Template cfDNA was derived from melanoma cell line culture media. The *CDKN2A* assays were performed using cfDNA from HDF1314-derived supernatant, while the remaining four assays were conducted from purified DNA from melanoma cell lines. (**B**) The maximum fold increase in DNA yield (copies/20 µL; post/pre-TOP-PCR) is shown for each target amplicon after 15 cycles of TOP-PCR amplification. This value represents the highest amplification yield observed for each target, highlighting the most effective enhancer (above each column) for each amplicon. The corresponding optimum amplification conditions for each DNA target is indicated: S—standard TOP-PCR conditions with no enhancer; B—TOP-PCR with 0.5 M betaine; D—TOP-PCR with 5% DMSO.

**Table 1 biomolecules-15-00883-t001:** Detection of multiple tumour-associated variants post-TOP-PCR amplification.

Patient	Progressed	Unamplified cfDNA Yield (ng)	Amplified cfDNA Yield (ng) ^a^	Mutation 1 Droplets ^b^	Mutation 2 Droplets ^b^
**42193**	Yes	68.7	501.6	0	7
**52318**	Yes	34.9	415.2	0	5
**58647**	Yes	54.5	423.6	0	8
**01622**	Yes	48.0	310.8	440	20
**38965**	Yes	135.3	612.0	16	30
**44691**	Yes	125.6	536.4	30	11
**50487**	Yes	65.4	436.8	110	48
**01523**	No	24.0	369.6	0	0
**01509**	No	21.0	418.8	0	0
**44894**	No	42.0	381.6	0	0
**28124**	No	61.1	308.4	0	0
**39346**	No	45.5	292.8	0	0

^a^ 20 ng input cfDNA was amplified for seven cycles. cfDNA yield (ng) was quantitated using Qubit High Sensitivity dsDNA Kit and Qubit 3.0 Fluorometer. 40 ng amplified cfDNA was used in each ddPCR. ^b^ Number of FAM+/HEX- droplets is shown. Mutations screened include *BRAF* c.1780G>A p.D594N, *NRAS* c.181C>A p.Q61K, *NRAS* c.182A>T p.Q61L, *NRAS* c.182A>G p.Q61R, *TERT* c.-146C>T, *IDH1* c.395G>A p.R132H, *TP53* c.659A>G p.Y220C, *BRAF* c.1798_1799delGTinsAA p.V600K, *RAC1* c.85C>T p.P29S, *BRAF* c.1790T>A p.L597Q, and *BRAF* c.1799T>A p.V600E.

## Data Availability

Data are available within the Article and Appendix A.

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
