# Peer review of "Pre-Amplification of Cell-Free DNA: Balancing Amplification Errors with Enhanced Sensitivity"

_biomolecules, 2025, doi:10.3390/biom15060883_

Round 1

Reviewer 1 Report

Comments and Suggestions for Authors

The authors investigated whether pre-amplifying cfDNA using polyA adaptors with low PCR cycles could enhance detection of oncology mutations by droplet digital PCR (ddPCR) by increasing ctDNA molecules. The study showed that using 20ng DNA with low preamplification cycles enables identification of ctDNA mutations difficult to detect with ddPCR on unamplified cfDNA. However, while preamplification enhances sensitivity, the authors demonstrated that amplification is neither uniform nor unbiased. Certain amplicons, due to DNA fragment quality, size distribution, GC content, and adaptor ligation variability, are more amplified than others, affecting ctDNA representation. Additionally, preamplification of cfDNA increases mutations, causing false positives. In a cost-benefit analysis, preamplification with TOP-PCR could be viable compared to molecular barcoding-based next-generation sequencing (NGS), offering good sensitivity and reliability at lower cost and time. However, the inability to homogeneously amplify cfDNA and accumulation of non-specific mutations due to PCR are significant limitations reducing test accuracy. The study was well-designed with appropriate controls. It cannot be definitively determined whether TOP-PCR is the best technique for pre-amplifying cfDNA, as no comparative study has conclusively demonstrated its benefits over other methods, despite mentions of less effective procedures in literature. Beyond these considerations, I have no further comments or experimental suggestions propose to the authors.

Reviewer 2 Report

Comments and Suggestions for Authors

Comments and Suggestions

The topic and approach are very timely and interesting. The major benefit of the TOP-PCR approach is then to have enough input for several dPCR, including different HotSpots and replicas. This helps identifiying patients in progress when only a part of HotSpots deliver enough positive copies, mainly due to different PCR efficiencies.

However, the paper seems to be a part of a already published paper (ref 21): Chan et al, 2024. In that paper, they look as well at melanoma stage III patients using the TOP-PCR approach and dPCR. The TOP-PCR dPCR protocol in that already published paper is as following:

T he DNA TOP-PCR cfDNA pre-amplification kit (Top Science Biotechnologies, Taiwan, Cat No. D01) was used to enhance ctDNA sensitivity [14]. Ligation and amplifi cation were performed as described by the manufacturer, except that input cfDNA was increased to 20ng cfDNA and amplified for only 5 cycles. The TOP-PCR pre-ampli f ication method was implemented for pre-treatment cfDNA samples with low/undetectable ctDNA (i.e. less than 10 ctDNA droplets detected via standard ddPCR), and for all post-surgery cfDNA samples. For pre-ampli f ication experiments, 5 µl of pre-amplified cfDNA (10 40ng) underwent three independent ddPCR experiments to validate ctDNA positivity.

So, taking these information my first question is:

  • Are the patients in the previous paper and now partially the same?
  • Are some of the experiments and results presented in both papers?

If so, I think this has to be outlined more clearly. As well it would be good that the paper mentions the paper from 2024 earlier in the story and not at the very end of the discussion at line 306.

Technically the paper is nicely set up and usefull for in-depth insight (d)dPCR analysis with low concentration isolates.

Some points would be interesting to add to the paper:

  • Figure 1: The approx TOP-PCR yield with 20ng input with 15 cycles and as well with 5 cycles is 500ng. The authors hypothesise that this may be due to saturation of reaction components. In Figure 1C the authors show increasing performance till cycle 7. It would be interesting to see at which cycle the saturation point occurs and starts to drop in efficiency.

  • Line 130: the authors state that they selected then 5 or 7 cycles for most of the analyses. Could the final selected cycles be included in the table of the used assay? What is meant with ‘most’? Are there any assays run then with different cycles?

  • Line 137 / Figure 2B / Line 278-281: do you have any explanation for the selective amplification of the Di-Nucleosome cfDNA? For which fragment length enrichment was the TOP-PCR optimized? If for fragmented gDNA and NGS it would make sense to optimize it in a range of 300-500bp; so maybe the protocol for cfDNA can be adapted?

  • Line 167 / Figure 2: cfDNA from 3 melanoma patients showed different yields after TOP-PCR. Do you have any explanation for that? Had this three isolates comparable stock concentrations? If not, the volume input into the TOP-PCR might have then influenced the PCR efficiency with any inhibitory component. Could you comment that?

  • Line 183 / in suppl table 3: size and GC content is nicely summarysed in the table. Why is the fact not discussed that the GC content in the TERT Promoter region is highest und thus the addition of betaine and DMSO improves dPCR efficiency? This is known from literature. Would it then be an option to chose the TOP-PCR conditions depeding on the targets you want to look at and then pool it again for dPCR analysis? If you run single-target dPCR approaches you of course do not need to pool the TOP-PCRs again but for high multiplexing in a 1-well format this might be worthy. Maybe you can include this in the discussion in lane 314?

  • Table 1: as far as I can see, the data was generated from 20ng input, 7 cycles, 40ng dPCR input, no replicas. This is somehow discrepant to the final setup with 5 cycles and a positive call when two independent ddPCRs had at least 2 positive copies (lane 285-289). This is then further complicated when looking at the data from suppl table 4, where 5 cycles were used. The message for me would then be either doing triplicats of unamplified input, duplicats of 5 cycle amp or 1-well analysis with 7 cycles. In the text reading flow it is thus not easy to follow when which conditions were chosen and why. Maybe this could be better structured or highlighted in subtitles.

Confusing to me is then why in the paper from 2024 a different setup was chosen (of course I cannot know whether all these experiments were done in parallel in the same time-setting, but this is now my assumption):

20ng cfDNA and amplified for only 5 cycles. The TOP-PCR pre-ampli f ication method was implemented for pre-treatment cfDNA samples with low/undetectable ctDNA (i.e. less than 10 ctDNA droplets detected via standard ddPCR), and for all post-surgery cfDNA samples. For pre-ampli f ication experiments, 5 µl of pre-amplified cfDNA (10 40ng) underwent three independent ddPCR experiments to validate ctDNA positivity.

Obviously, there was an approach selected with triplicats but without mentioning how many positive replicas and how many positive copies have to be detected. I wonder how the paper results there would have been when the workflow and guidelines from the paper presented here and described in suppl. Figure 3 were chosen.

All in all, the experimental setups are not streamlined throughout the paper which makes it hard to read an interpret.

  • Lane 242: It would be interesting to see a TapeStation Histogram from cfDNA supernatant derived from cultured HDF1314 human dermal fibroblast. Following all the experimental setup, it would as well be interesting to see how this isolate performs with small molecule additives with the targets used in Figure 3. This leads me to an additional question: which input from which isolates were used in the efficiency experiment lane 175ff? Probably it was already the fibroblast supernatant?

  • Lane 250: The authors mention 20 tested mutations in suppl. Fig 2. Could you specify these assays or mark them in suppl table 5? Obviously the assays are not the same, because in suppl. Table 5 only 18 assays are listed. From the results presented in suppl table 5, I wonder why the authors went further with any 7 cycle testing at all, because an increase of 33% FP with a range of 1-6 positive copies is in my eyes too high. This is one major concern for me, since the story line of the paper is build up from table 1 with patient data, where the setup with 7 cycles was used.

Reviewer 3 Report

Comments and Suggestions for Authors

This work has investigated the clinical value of TOP-PCR pre-amplification for ctDNA analysis. The research indicate that the approach may offer enhanced ctDNA detection sensitivity which is valuable in clinical practice, while errors introduced during pre-amplification can compromise specificity. Some revisions are necessary:

1) Figure 1, what’s the reason for the inverse correlation using 15 cycles (recommended by the manufacturer), while it changes to normal relationship at 5 cycles?

2) Figure 2a, can the authors briefly explain the reason for a band at 35 bp in Sample 1 and 3, but absent in Sample 2?

3) Can the authors clearly compare the false positive and/or false negative values by different methods?

4) Figure 3b, I cannot understand why the authors use a term “the maximum fold increase in DNA yield” obtained by different PCR methods in a graph. To compare the performances by different methods, the fold increase should be listed separately.

5) The authors should check the whole manuscript to avoid unnecessary “-” in words such as “applica-tions”, “Ag-ilent”.

Round 2

Reviewer 2 Report

Comments and Suggestions for Authors

Dear authors

thanks for the clarifications. The paper gives now very clear and important technical insight into the (dPCR) MRD problematic.

Minor spelling error: 308ff: ....a true-positive mutation(s), and the inclusion of (at) least two independent...
